# Variational Information Maximization for Feature Selection

**Shuyang Gao**          **Greg Ver Steeg**          **Aram Galstyan**
University of Southern California, Information Sciences Institute
gaos@usc.edu, gregv@isi.edu, galstyan@isi.edu

## Abstract

Feature selection is one of the most fundamental problems in machine learning. An extensive body of work on information-theoretic feature selection exists which is based on maximizing mutual information between subsets of features and class labels. Practical methods are forced to rely on approximations due to the difficulty of estimating mutual information. We demonstrate that approximations made by existing methods are based on unrealistic assumptions. We formulate a more flexible and general class of assumptions based on variational distributions and use them to tractably generate lower bounds for mutual information. These bounds define a novel information-theoretic framework for feature selection, which we prove to be optimal under tree graphical models with proper choice of variational distributions. Our experiments demonstrate that the proposed method strongly outperforms existing information-theoretic feature selection approaches.

## 1   Introduction

Feature selection is one of the fundamental problems in machine learning research [1, 2]. Its problematic issues include a large number of features that are either *irrelevant* or *redundant* for the task at hand. In these cases, it is often advantageous to pick a smaller subset of features to avoid over-fitting, to speed up computation, or simply to improve the interpretability of the results.

Feature selection approaches are usually categorized into three groups: *wrapper*, *embedded* and *filter* [3, 4, 5]. The first two methods, *wrapper* and *embedded*, are considered *classifier-dependent*, i.e., the selection of features somehow depends on the classifier being used. *Filter* methods, on the other hand, are *classifier-independent* and define a scoring function between features and labels in the selection process.

Because *filter* methods may be employed in conjunction with a wide variety of classifiers, it is important that the scoring function of these methods is as general as possible. Since mutual information (MI) is a general measure of dependence with several unique properties [6], many MI-based scoring functions have been proposed as *filter* methods [7, 8, 9, 10, 11, 12]; see [5] for an exhaustive list.

Owing to the difficulty of estimating mutual information in high dimensions, most existing MI-based feature selection methods are based on various low-order approximations for mutual information. While those approximations have been successful in certain applications, they are heuristic in nature and lack theoretical guarantees. In fact, as we demonstrate in Sec. 2.2, a large family of approximate methods are based on two assumptions that are mutually inconsistent.

To address the above shortcomings, in this paper we introduce a novel feature selection method based on a variational lower bound on mutual information; a similar bound was previously studied within the Infomax learning framework [13]. We show that instead of maximizing the mutual information, which is intractable in high dimensions (hence the introduction of many heuristics), we can

maximize a lower bound on the MI with the proper choice of tractable variational distributions. We use this lower bound to define an objective function and derive a forward feature selection algorithm.

We provide a rigorous proof that the forward feature selection is optimal under tree graphical models by choosing an appropriate variational distribution. This is in contrast with previous information-theoretic feature selection methods which lack any performance guarantees. We also conduct empirical validation on various datasets and demonstrate that the proposed approach outperforms state-of-the-art information-theoretic feature selection methods.

In Sec. 2 we introduce general MI-based feature selection methods and discuss their limitations. Sec. 3 introduces the variational lower bound on mutual information and proposes two specific variational distributions. In Sec. 4, we report results from our experiments, and compare the proposed approach with existing methods.

## 2   Information-Theoretic Feature Selection Background

### 2.1   Mutual Information-Based Feature Selection

Consider a supervised learning scenario where $\mathbf{x} = \{\mathbf{x}_1, \mathbf{x}_2, ..., \mathbf{x}_D\}$ is a $D$-dimensional input feature vector, and $\mathbf{y}$ is the output label. In *filter* methods, the mutual information-based feature selection task is to select $T$ features $\mathbf{x}_{S^*} = \{\mathbf{x}_{f_1}, \mathbf{x}_{f_2}, ..., \mathbf{x}_{f_T}\}$ such that the mutual information between $\mathbf{x}_{S^*}$ and $\mathbf{y}$ is maximized. Formally,

$$S^* = \arg\max_{S} I\left(\mathbf{x}_S : \mathbf{y}\right) \quad s.t. \ |S| = T \tag{1}$$

where $I(\cdot)$ denotes the mutual information [6].

**Forward Sequential Feature Selection**     Maximizing the objective function in Eq. 1 is generally NP-hard. Many MI-based feature selection methods adopt a greedy method, where features are selected incrementally, one feature at a time. Let $S^{t-1} = \{\mathbf{x}_{f_1}, \mathbf{x}_{f_2}, ..., \mathbf{x}_{f_{t-1}}\}$ be the selected feature set after time step $t-1$. According to the greedy method, the next feature $f_t$ at step $t$ is selected such that

$$f_t = \arg\max_{i \notin S^{t-1}} I\left(\mathbf{x}_{S^{t-1} \cup i} : \mathbf{y}\right) \tag{2}$$

where $\mathbf{x}_{S^{t-1} \cup i}$ denotes $\mathbf{x}$'s projection into the feature space $S^{t-1} \cup i$. As shown in [5], the mutual information term in Eq. 2 can be decomposed as:

$$
\begin{aligned}
I\left(\mathbf{x}_{S^{t-1} \cup i} : \mathbf{y}\right) &= I\left(\mathbf{x}_{S^{t-1}} : \mathbf{y}\right) + I\left(\mathbf{x}_i : \mathbf{y} | \mathbf{x}_{S^{t-1}}\right) \\
&= I\left(\mathbf{x}_{S^{t-1}} : \mathbf{y}\right) + I\left(\mathbf{x}_i : \mathbf{y}\right) - I\left(\mathbf{x}_i : \mathbf{x}_{S^{t-1}}\right) + I\left(\mathbf{x}_i : \mathbf{x}_{S^{t-1}} | \mathbf{y}\right) \\
&= I\left(\mathbf{x}_{S^{t-1}} : \mathbf{y}\right) + I\left(\mathbf{x}_i : \mathbf{y}\right) \\
&\quad - \left(H\left(\mathbf{x}_{S^{t-1}}\right) - H\left(\mathbf{x}_{S^{t-1}} | \mathbf{x}_i\right)\right) + \left(H\left(\mathbf{x}_{S^{t-1}} | \mathbf{y}\right) - H\left(\mathbf{x}_{S^{t-1}} | \mathbf{x}_i, \mathbf{y}\right)\right)
\end{aligned}
\tag{3}
$$

where $H(\cdot)$ denotes the entropy [6]. Omitting the terms that do not depend on $\mathbf{x}_i$ in Eq. 3, we can rewrite Eq. 2 as follows:

$$f_t = \arg\max_{i \notin S^{t-1}} I\left(\mathbf{x}_i : \mathbf{y}\right) + H\left(\mathbf{x}_{S^{t-1}} | \mathbf{x}_i\right) - H\left(\mathbf{x}_{S^{t-1}} | \mathbf{x}_i, \mathbf{y}\right) \tag{4}$$

The greedy learning algorithm has been analyzed in [14].

### 2.2   Limitations of Previous MI-Based Feature Selection Methods

Estimating high-dimensional information-theoretic quantities is a difficult task.     Therefore, most MI-based feature selection methods propose low-order approximation to $H\left(\mathbf{x}_{S^{t-1}} | \mathbf{x}_i\right)$ and $H\left(\mathbf{x}_{S^{t-1}} | \mathbf{x}_i, \mathbf{y}\right)$ in Eq. 4. A general family of methods rely on the following approximations [5]:

$$
\begin{aligned}
H\left(\mathbf{x}_{S^{t-1}} | \mathbf{x}_i\right) &\approx \sum_{k=1}^{t-1} H\left(\mathbf{x}_{f_k} | \mathbf{x}_i\right) \\
H\left(\mathbf{x}_{S^{t-1}} | \mathbf{x}_i, \mathbf{y}\right) &\approx \sum_{k=1}^{t-1} H\left(\mathbf{x}_{f_k} | \mathbf{x}_i, \mathbf{y}\right)
\end{aligned}
\tag{5}
$$

The approximations in Eq. 5 become exact under the following two assumptions [5]:

*Assumption 1. (Feature Independence Assumption)* $p\left(\mathbf{x}_{S^{t-1}}|\mathbf{x}_i\right) = \prod\limits_{k=1}^{t-1} p\left(\mathbf{x}_{f_k}|\mathbf{x}_i\right)$

*Assumption 2. (Class-Conditioned Independence Assumption)* $p\left(\mathbf{x}_{S^{t-1}}|\mathbf{x}_i,\mathbf{y}\right) = \prod\limits_{k=1}^{t-1} p\left(\mathbf{x}_{f_k}|\mathbf{x}_i,\mathbf{y}\right)$

*Assumption 1* and *Assumption 2* mean that the selected features are independent and class-conditionally independent, respectively, given the unselected feature $\mathbf{x}_i$ under consideration.

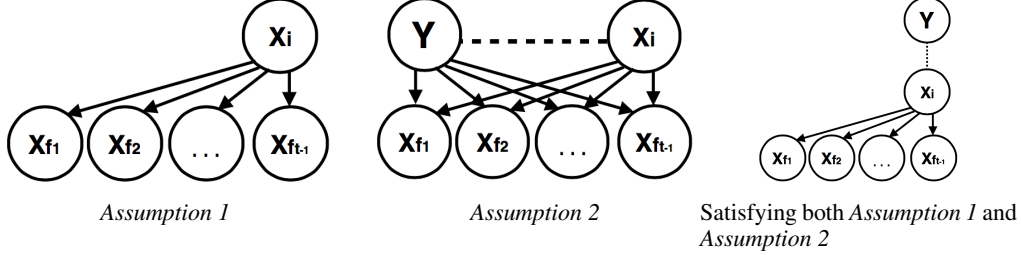

Figure 1: The first two graphical models show the assumptions of traditional MI-based feature selection methods. The third graphical model shows a scenario when both *Assumption 1* and *Assumption 2* are true. Dashed line indicates there may or may not be a correlation between two variables.

We now demonstrate that the two assumptions cannot be valid simultaneously unless the data has a very specific (and unrealistic) structure. Indeed, consider the graphical models consistent with either assumption, as illustrated in Fig. 1. If *Assumption 1* holds true, then $\mathbf{x}_i$ is the only common cause of the previously selected features $S^{t-1} = \{\mathbf{x}_{f_1}, \mathbf{x}_{f_2}, ..., \mathbf{x}_{f_{t-1}}\}$, so that those features become independent when conditioned on $\mathbf{x}_i$. On the other hand, if *Assumption 2* holds, then the features depend both on $\mathbf{x}_i$ and class label $\mathbf{y}$; therefore, generally speaking, distribution over those features does not factorize by solely conditioning on $\mathbf{x}_i$—there will be remnant dependencies due to $\mathbf{y}$. Thus, if *Assumption 2* is true, then *Assumption 1* cannot be true in general, unless the data is generated according to a very specific model shown in the rightmost model in Fig. 1. Note, however, that in this case, $\mathbf{x}_i$ becomes the most important feature because $I(\mathbf{x}_i : \mathbf{y}) > I(\mathbf{x}_{S^{t-1}} : \mathbf{y})$; then we should have selected $\mathbf{x}_i$ at the very first step, contradicting the feature selection process.

As we mentioned above, most existing methods implicitly or explicitly adopt both assumptions or their stronger versions, as shown in [5]—including mutual information maximization (MIM) [15], joint mutual information (JMI) [8], conditional mutual information maximization (CMIM) [9], maximum relevance minimum redundancy (mRMR) [10], conditional Infomax feature extraction (CIFE) [16], etc. Approaches based on global optimization of mutual information, such as quadratic programming feature selection ($\mathcal{QPFS}$) [11] and the state-of-the-art conditional mutual information-based spectral method ($\mathcal{SPEC}_{\mathcal{CMI}}$) [12], are derived from the previous greedy methods and therefore also implicitly rely on those two assumptions.

In the next section we address these issues by introducing a novel information-theoretic framework for feature selection. Instead of estimating mutual information and making mutually inconsistent assumptions, our framework formulates a tractable variational lower bound on mutual information, which allows a more flexible and general class of assumptions via appropriate choices of variational distributions.

## 3 Method

### 3.1 Variational Mutual Information Lower Bound

Let $p(\mathbf{x}, \mathbf{y})$ be the joint distribution of input ($\mathbf{x}$) and output ($\mathbf{y}$) variables. Barber & Agkov [13] derived the following lower bound for mutual information $I(\mathbf{x} : \mathbf{y})$ by using the non-negativity of KL-divergence, i.e., $\sum_{\mathbf{x}} p\left(\mathbf{x}|\mathbf{y}\right) \log \frac{p(\mathbf{x}|\mathbf{y})}{q(\mathbf{x}|\mathbf{y})} \geq 0$ gives:

$$I\left(\mathbf{x} : \mathbf{y}\right) \geq H\left(\mathbf{x}\right) + \langle \ln q\left(\mathbf{x}|\mathbf{y}\right)\rangle_{p(\mathbf{x},\mathbf{y})} \tag{6}$$

where angled brackets represent averages and $q(\mathbf{x}|\mathbf{y})$ is an arbitrary variational distribution. This bound becomes exact if $q(\mathbf{x}|\mathbf{y}) \equiv p(\mathbf{x}|\mathbf{y})$.

It is worthwhile to note that in the context of unsupervised representation learning, $p(\mathbf{y}|\mathbf{x})$ and $q(\mathbf{x}|\mathbf{y})$ can be viewed as an *encoder* and a *decoder*, respectively. In this case, $\mathbf{y}$ needs to be learned by maximizing the lower bound in Eq. 6 by iteratively adjusting the parameters of the encoder and decoder, such as [13, 17].

## 3.2 Variational Information Maximization for Feature Selection

Naturally, in terms of information-theoretic feature selection, we could also try to optimize the variational lower bound in Eq. 6 by choosing a subset of features $S^*$ in $\mathbf{x}$, such that,

$$S^* = \arg\max_S \left\{ H\left(\mathbf{x}_S\right) + \langle \ln q\left(\mathbf{x}_S|\mathbf{y}\right) \rangle_{p(\mathbf{x}_S, \mathbf{y})} \right\} \tag{7}$$

However, the $H(\mathbf{x}_S)$ term in RHS of Eq. 7 is still intractable when $\mathbf{x}_S$ is very high-dimensional.

Nonetheless, by noticing that variable $\mathbf{y}$ is the class label, which is usually discrete, and hence $H(\mathbf{y})$ is fixed and tractable, by symmetry we switch $\mathbf{x}$ and $\mathbf{y}$ in Eq. 6 and rewrite the lower bound as follows:

$$I\left(\mathbf{x}:\mathbf{y}\right) \geq H\left(\mathbf{y}\right) + \langle \ln q\left(\mathbf{y}|\mathbf{x}\right) \rangle_{p(\mathbf{x}, \mathbf{y})} = \left\langle \ln\left(\frac{q\left(\mathbf{y}|\mathbf{x}\right)}{p\left(\mathbf{y}\right)}\right) \right\rangle_{p(\mathbf{x}, \mathbf{y})} \tag{8}$$

The equality in Eq. 8 is obtained by noticing that $H(\mathbf{y}) = \langle -\ln p\left(\mathbf{y}\right) \rangle_{p(\mathbf{y})}$.

By using Eq. 8, the lower bound optimal subset $S^*$ of $\mathbf{x}$ becomes:

$$S^* = \arg\max_S \left\{ \left\langle \ln\left(\frac{q\left(\mathbf{y}|\mathbf{x}_S\right)}{p\left(\mathbf{y}\right)}\right) \right\rangle_{p(\mathbf{x}_S, \mathbf{y})} \right\} \tag{9}$$

### 3.2.1 Choice of Variational Distribution

$q(\mathbf{y}|\mathbf{x}_S)$ in Eq. 9 can be *any* distribution as long as it is normalized. We need to choose $q(\mathbf{y}|\mathbf{x}_S)$ to be as general as possible while still keeping the term $\langle \ln q\left(\mathbf{y}|\mathbf{x}_S\right) \rangle_{p(\mathbf{x}_S, \mathbf{y})}$ tractable in Eq. 9.

As a result, we set $q(\mathbf{y}|\mathbf{x}_S)$ as

$$q\left(\mathbf{y}|\mathbf{x}_S\right) = \frac{q\left(\mathbf{x}_S, \mathbf{y}\right)}{q\left(\mathbf{x}_S\right)} = \frac{q\left(\mathbf{x}_S|\mathbf{y}\right) p\left(\mathbf{y}\right)}{\sum_{\mathbf{y}'} q\left(\mathbf{x}_S|\mathbf{y}'\right) p\left(\mathbf{y}'\right)} \tag{10}$$

We can verify that Eq. 10 is normalized even if $q(\mathbf{x}_S|\mathbf{y})$ is not normalized.

If we further denote,

$$q\left(\mathbf{x}_S\right) = \sum_{\mathbf{y}'} q\left(\mathbf{x}_S|\mathbf{y}'\right) p\left(\mathbf{y}'\right) \tag{11}$$

then by combining Eqs. 9 and 10, we get,

$$I\left(\mathbf{x}_S:\mathbf{y}\right) \geq \left\langle \ln\left(\frac{q\left(\mathbf{x}_S|\mathbf{y}\right)}{q\left(\mathbf{x}_S\right)}\right) \right\rangle_{p(\mathbf{x}_S, \mathbf{y})} \equiv I_{LB}\left(\mathbf{x}_S:\mathbf{y}\right) \tag{12}$$

And we also have the following equation which shows the gap between $I(\mathbf{x}_S:\mathbf{y})$ and $I_{LB}(\mathbf{x}_S:\mathbf{y})$,

$$I\left(\mathbf{x}_S:\mathbf{y}\right) - I_{LB}\left(\mathbf{x}_S:\mathbf{y}\right) = \langle KL\left(p\left(\mathbf{y}|\mathbf{x}_S\right) || q\left(\mathbf{y}|\mathbf{x}_S\right)\right) \rangle_{p(\mathbf{x}_S)} \tag{13}$$

**Auto-Regressive Decomposition.** Now that $q(\mathbf{y}|\mathbf{x}_S)$ is defined, all we need to do is model $q(\mathbf{x}_S|\mathbf{y})$ under Eq. 10, and $q(\mathbf{x}_S)$ is easy to compute based on $q(\mathbf{x}_S|\mathbf{y})$. Here we decompose $q(\mathbf{x}_S|\mathbf{y})$ as an auto-regressive distribution assuming $T$ features in $S$:

$$q\left(\mathbf{x}_S|\mathbf{y}\right) = q\left(\mathbf{x}_{f_1}|\mathbf{y}\right) \prod_{t=2}^{T} q\left(\mathbf{x}_{f_t}|\mathbf{x}_{f_{<t}}, \mathbf{y}\right) \tag{14}$$

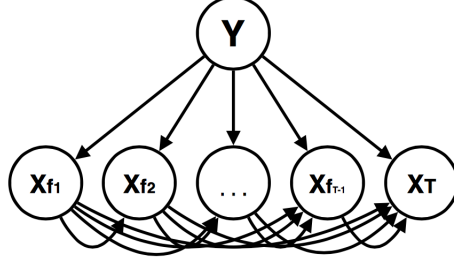

Figure 2: Auto-regressive decomposition for $q(\mathbf{x}_S|\mathbf{y})$

where $\mathbf{x}_{f_{<t}}$ denotes $\{\mathbf{x}_{f_1}, \mathbf{x}_{f_2}, ..., \mathbf{x}_{f_{t-1}}\}$. The graphical model in Fig. 2 demonstrates this decomposition. The main advantage of this model is that it is well-suited for the forward feature selection procedure where one feature is selected at a time (which we will explain in Sec. 3.2.3). And if $q\left(\mathbf{x}_{f_t}|\mathbf{x}_{f_{<t}}, \mathbf{y}\right)$ is tractable, then so is the whole distribution $q(\mathbf{x}_S|\mathbf{y})$. Therefore, we would find tractable $Q$-distributions over $q\left(\mathbf{x}_{f_t}|\mathbf{x}_{f_{<t}}, \mathbf{y}\right)$. Below we illustrate two such $Q$-distributions.

**Naive Bayes $Q$-distribution.** A natural idea would be to assume $\mathbf{x}_t$ is independent of other variables given $\mathbf{y}$, i.e.,

$$q\left(\mathbf{x}_{f_t}|\mathbf{x}_{f_{<t}}, \mathbf{y}\right) = p\left(\mathbf{x}_{f_t}|\mathbf{y}\right) \tag{15}$$

Then the variational distribution $q(\mathbf{y}|\mathbf{x}_S)$ can be written based on Eqs. 10 and 15 as follows:

$$q\left(\mathbf{y}|\mathbf{x}_S\right) = \frac{p\left(\mathbf{y}\right) \prod\limits_{j \in S} p\left(\mathbf{x}_j|\mathbf{y}\right)}{\sum\limits_{\mathbf{y}'} p\left(\mathbf{y}'\right) \prod\limits_{j \in S} p\left(\mathbf{x}_j|\mathbf{y}'\right)} \tag{16}$$

And we also have the following theorem:

**Theorem 3.1** (Exact Naive Bayes). *Under Eq. 16, the lower bound in Eq. 8 becomes exact if and only if data is generated by a Naive Bayes model, i.e., $p\left(\mathbf{x}, \mathbf{y}\right) = p\left(\mathbf{y}\right) \prod\limits_i p\left(\mathbf{x}_i|\mathbf{y}\right)$.*

The proof for Theorem 3.1 becomes obvious by using the mutual information definition. Note that the most-cited MI-based feature selection method mRMR [10] also assumes conditional independence given the class label $\mathbf{y}$ as shown in [5, 18, 19], but they make additional stronger independence assumptions among only feature variables.

**Pairwise $Q$-distribution.** We now consider an alternative approach that is more general than the Naive Bayes distribution:

$$q\left(\mathbf{x}_{f_t}|\mathbf{x}_{f_{<t}}, \mathbf{y}\right) = \left(\prod_{i=1}^{t-1} p\left(\mathbf{x}_{f_t}|\mathbf{x}_{f_i}, \mathbf{y}\right)\right)^{\frac{1}{t-1}} \tag{17}$$

In Eq. 17, we assume $q\left(\mathbf{x}_{f_t}|\mathbf{x}_{f_{<t}}, \mathbf{y}\right)$ to be the geometric mean of conditional distributions $q(\mathbf{x}_{f_t}|\mathbf{x}_{f_i}, \mathbf{y})$. This assumption is tractable as well as reasonable because if the data is generated by a Naive Bayes model, the lower bound in Eq. 8 also becomes exact using Eq. 17 due to $p\left(\mathbf{x}_{f_t}|\mathbf{x}_{f_i}, \mathbf{y}\right) \equiv p\left(\mathbf{x}_{f_t}|\mathbf{y}\right)$ in that case.

### 3.2.2 Estimating Lower Bound From Data

Assuming either Naive Bayes $Q$-distribution or pairwise $Q$-distribution, it is convenient to estimate $q(\mathbf{x}_S|\mathbf{y})$ and $q(\mathbf{x}_S)$ in Eq. 12 by using plug-in probability estimators for discrete data or one/two-dimensional density estimators for continuous data. We also use the sample mean to approximate the expectation term in Eq. 12. Our final estimator for $I_{LB}\left(\mathbf{x}_S : \mathbf{y}\right)$ is written as follows:

$$\widehat{I}_{LB}\left(\mathbf{x}_S : \mathbf{y}\right) = \frac{1}{N} \sum_{\mathbf{x}^{(k)}, \mathbf{y}^{(k)}} \ln \frac{\widehat{q}\left(\mathbf{x}_S^{(k)}|\mathbf{y}^{(k)}\right)}{\widehat{q}\left(\mathbf{x}_S^{(k)}\right)} \tag{18}$$

where $\left\{\mathbf{x}^{(k)}, \mathbf{y}^{(k)}\right\}$ are samples from data, and $\widehat{q}(\cdot)$ denotes the estimate for $q(\cdot)$.

### 3.2.3 Variational Forward Feature Selection Under Auto-Regressive Decomposition

After defining $q(\mathbf{y}|\mathbf{x}_S)$ in Eq. 10 and auto-regressive decomposition of $q(\mathbf{x}_S|\mathbf{y})$ in Eq. 15, we are able to do the forward feature selection previously described in Eq. 2, but replace the mutual information with its lower bound $\widehat{I}_{LB}$. Recall that $S^{t-1}$ is the set of selected features after step $t-1$, then the feature $f_t$ will be selected at step $t$ such that

$$f_t = \underset{i \notin S^{t-1}}{\arg\max}\, \widehat{I}_{LB}\left(\mathbf{x}_{S^{t-1} \cup i} : \mathbf{y}\right) \tag{19}$$

where $\widehat{I}_{LB}\left(\mathbf{x}_{S^{t-1} \cup i} : \mathbf{y}\right)$ can be obtained from $\widehat{I}_{LB}\left(\mathbf{x}_{S^{t-1}} : \mathbf{y}\right)$ recursively by auto-regressive decomposition $q\left(\mathbf{x}_{S^{t-1} i}|\mathbf{y}\right) = q\left(\mathbf{x}_{S^{t-1}}|\mathbf{y}\right) q\left(\mathbf{x}_i|\mathbf{x}_{S^{t-1}}, \mathbf{y}\right)$ where $q\left(\mathbf{x}_{S^{t-1}}|\mathbf{y}\right)$ is stored at step $t-1$.

This forward feature selection can be done under auto-regressive decomposition in Eqs. 10 and 14 for *any* $Q$-distribution. However, calculating $q(\mathbf{x}_i|\mathbf{x}_{S^t}, \mathbf{y})$ may vary according to different $Q$-distributions. We can verify that it is easy to get $q(\mathbf{x}_i|\mathbf{x}_{S^t}, \mathbf{y})$ recursively from $q(\mathbf{x}_i|\mathbf{x}_{S^{t-1}}, \mathbf{y})$ under Naive Bayes or pairwise $Q$-distribution. We call our algorithm under these two $Q$-distributions $\mathcal{VMI}_{naive}$ and $\mathcal{VMI}_{pairwise}$ respectively.

It is worthwhile noting that the lower bound does not always increase at each step. A decrease in lower bound at step $t$ indicates that the $Q$-distribution would approximate the underlying distribution worse than it did at previous step $t-1$. In this case, the algorithm would re-maximize the lower bound from zero with only the remaining unselected features. We summarize the concrete implementation of our algorithms in supplementary Sec. A.

**Time Complexity.** Although our algorithm needs to calculate the distributions at each step, we only need to calculate the probability value at each sample point. For both $\mathcal{VMI}_{naive}$ and $\mathcal{VMI}_{pairwise}$, the total computational complexity is $O(NDT)$ assuming $N$ as number of samples, $D$ as total number of features, $T$ as number of final selected features. The detailed time analysis is left for the supplementary Sec. A. As shown in Table 1, our methods $\mathcal{VMI}_{naive}$ and $\mathcal{VMI}_{pairwise}$ have the same time complexity as mRMR [10], while the state-of-the-art global optimization method $\mathcal{SPEC}_{CMI}$ [12] is required to precompute the pairwise mutual information matrix, which gives a time complexity of $O(ND^2)$.

Table 1: **Time complexity in number of features $D$, selected number of features $d$, and number of samples $N$**

| Method | mRMR | $\mathcal{VMI}_{naive}$ | $\mathcal{VMI}_{pairwise}$ | $\mathcal{SPEC}_{CMI}$ |
|---|---|---|---|---|
| **Complexity** | $O(NDT)$ | $O(NDT)$ | $O(NDT)$ | $O(ND^2)$ |

**Optimality Under Tree Graphical Models.** Although our method $\mathcal{VMI}_{naive}$ assumes a Naive Bayes model, we can prove that this method is still optimal if the data is generated according to tree graphical models. Indeed, both of our methods, $\mathcal{VMI}_{naive}$ and $\mathcal{VMI}_{pairwise}$, will always prioritize the first layer features, as shown in Fig. 3. This optimality is summarized in Theorem B.1 in supplementary Sec. B.

## 4 Experiments

**Synthetic Data.** We begin with the experiments on a synthetic model according to the tree structure illustrated in the left part of Fig. 3. The detailed data generating process is shown in supplementary section D. The root node $\mathbf{Y}$ is a binary variable, while other variables are continuous. We use $\mathcal{VMI}_{naive}$ to optimize the lower bound $I_{LB}(\mathbf{x} : \mathbf{y})$. 5000 samples are used to generate the synthethic data, and variational $Q$-distributions are estimated by the kernel density estimator. We can see from the plot in the right-hand part of Fig. 3 that our algorithm, $\mathcal{VMI}_{naive}$, selects $\mathbf{x}_1$, $\mathbf{x}_2$, $\mathbf{x}_3$ as the first three features, although $\mathbf{x}_2$ and $\mathbf{x}_3$ are only weakly correlated with $\mathbf{y}$. If we continue to add deeper level features $\{\mathbf{x}_4, ..., \mathbf{x}_9\}$, the lower bound will decrease. For comparison, we also illustrate the mutual information between each single feature $\mathbf{x}_i$ and $\mathbf{y}$ in Table 2. We can see from Table 2 that it would choose $\mathbf{x}_1$, $\mathbf{x}_4$ and $\mathbf{x}_5$ as the top three features by using the maximum relevance criteria [15].

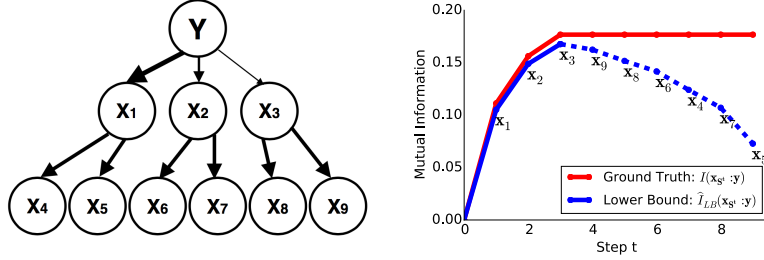

Figure 3: (Left) This is the generative model used for synthetic experiments. Edge thickness represents the relationship strength. (Right) Optimizing the lower bound by $\mathcal{VMI}_{naive}$. Variables under the blue line denote the features selected at each step. Dotted blue line shows the decreasing lower bound if adding more features. Ground-truth mutual information is obtained using $N = 100,000$ samples.

| feature$_i$ | $\mathbf{x}_1$ | $\mathbf{x}_2$ | $\mathbf{x}_3$ | $\mathbf{x}_4$ | $\mathbf{x}_5$ | $\mathbf{x}_6$ | $\mathbf{x}_7$ | $\mathbf{x}_8$ | $\mathbf{x}_9$ |
|---|---|---|---|---|---|---|---|---|---|
| $I(\mathbf{x}_i : \mathbf{y})$ | **0.111** | 0.052 | 0.022 | **0.058** | **0.058** | 0.025 | 0.029 | 0.012 | 0.013 |

Table 2: Mutual information between label $\mathbf{y}$ and each feature $\mathbf{x}_i$ for Fig. 3. $I(\mathbf{x}_i : \mathbf{y})$ is estimated using N=100,000 samples. Top three variables with highest mutual information are highlighted in bold.

**Real-World Data.** We compare our algorithms $\mathcal{VMI}_{naive}$ and $\mathcal{VMI}_{pairwise}$ with other popular information-theoretic feature selection methods, including mRMR [10], JMI [8], MIM [15], CMIM [9], CIFE [16], and $\mathcal{SPEC}_{\mathcal{CMI}}$ [12]. We use 17 well-known datasets in previous feature selection studies [5, 12] (all data are discretized). The dataset summaries are illustrated in supplementary Sec. C. We use the average cross-validation error rate on the range of 10 to 100 features to compare different algorithms under the same setting as [12]. Tenfold cross-validation is employed for datasets with number of samples $N \geq 100$ and leave-one-out cross-validation otherwise. The 3-nearest-neighbor classifier is used for Gisette and Madelon, following [5]. For the remaining datasets, the chosen classifier is Linear SVM, following [11, 12].

The experimental results can be seen in Table 3.[1] The entries with $*$ and $**$ indicate the best performance and the second best performance, respectively (in terms of average error rate). We also use the paired t-test at 5% significant level to test the hypothesis that $\mathcal{VMI}_{naive}$ or $\mathcal{VMI}_{pairwise}$ perform significantly better than other methods, or vice visa. Overall, we find that both of our methods, $\mathcal{VMI}_{naive}$ and $\mathcal{VMI}_{pairwise}$, strongly outperform other methods. This indicates that our variational feature selection framework is a promising addition to the current literature of information-theoretic feature selection.

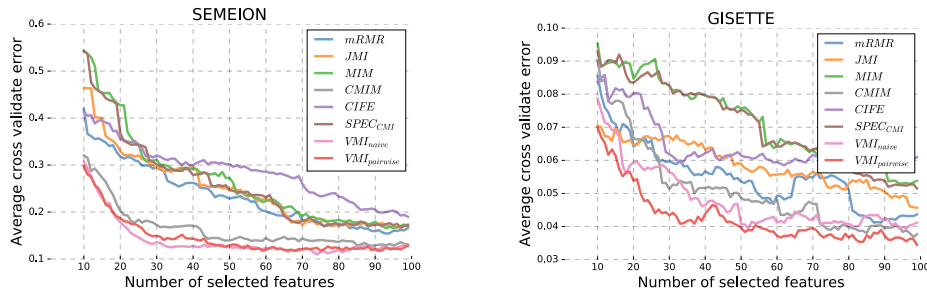

Figure 4: Number of selected features versus average cross-validation error in datasets Semeion and Gisette.

Table 3: **Average cross-validation error rate comparison of $\mathcal{VMI}$ against other methods. The last two lines indicate win(W)/tie (T)/ loss(L) for $\mathcal{VMI}_{naive}$ and $\mathcal{VMI}_{pairwise}$ respectively.**

| Dataset | mRMR | JMI | CMIM | $\mathcal{SPEC}_{\mathcal{CMI}}$ | $\mathcal{VMI}_{naive}$ | $\mathcal{VMI}_{pairwise}$ |
|---|---|---|---|---|---|---|
| Lung | **10.9±(4.7)**\*\* | 11.6±(4.7) | 11.4±(3.0) | 11.6±(5.6) | **7.4±(3.6)**\* | 14.5±(6.0) |
| Colon | 19.7±(2.6) | 17.3±(3.0) | 18.4±(2.6) | 16.1±(2.0) | **11.2±(2.7)**\* | **11.9±(1.7)**\*\* |
| Leukemia | 0.4±(0.7) | 1.4±(1.2) | 1.1±(2.0) | 1.8±(1.3) | **0.0±(0.1)**\* | **0.2±(0.5)**\*\* |
| Lymphoma | 5.6±(2.8) | 6.6±(2.2) | 8.6±(3.3) | 12.0±(6.6) | **3.7±(1.9)**\* | **5.2±(3.1)**\*\* |
| Splice | **13.6±(0.4)**\* | **13.7±(0.5)**\*\* | 14.7±(0.3) | **13.7±(0.5)**\*\* | **13.7±(0.5)**\*\* | **13.7±(0.5)**\*\* |
| Landsat | 19.5±(1.2) | 18.9±(1.0) | 19.1±(1.1) | 21.0±(3.5) | **18.8±(0.8)**\* | **18.8±(1.0)**\*\* |
| Waveform | **15.9±(0.5)**\* | **15.9±(0.5)**\* | 16.0±(0.7) | **15.9±(0.6)**\*\* | **15.9±(0.6)**\*\* | **15.9±(0.5)**\* |
| KrVsKp | **5.1±(0.7)**\*\* | 5.2±(0.6) | 5.3±(0.5) | **5.1±(0.6)**\* | 5.3±(0.5) | **5.1±(0.7)**\*\* |
| Ionosphere | 12.8±(0.9) | 16.6±(1.6) | 13.1±(0.8) | 16.8±(1.6) | **12.7±(1.9)**\*\* | **12.0±(1.0)**\* |
| Semeion | 23.4±(6.5) | 24.8±(7.6) | 16.3±(4.4) | 26.0±(9.3) | **14.0±(4.0)**\* | **14.5±(3.9)**\*\* |
| Multifeat. | 4.0±(1.6) | 4.0±(1.6) | 3.6±(1.2) | 4.8±(3.0) | **3.0±(1.1)**\* | **3.5±(1.1)**\*\* |
| Optdigits | 7.6±(3.3) | 7.6±(3.2) | **7.5±(3.4)**\*\* | 9.2±(6.0) | **7.2±(2.5)**\* | 7.6±(3.6) |
| Musk2 | **12.4±(0.7)**\* | 12.8±(0.7) | 13.0±(1.0) | 15.1±(1.8) | 12.8±(0.6) | **12.6±(0.5)**\*\* |
| Spambase | 6.9±(0.7) | 7.0±(0.8) | **6.8±(0.7)**\*\* | 9.0±(2.3) | **6.6±(0.3)**\* | **6.6±(0.3)**\* |
| Promoter | 21.5±(2.8) | 22.4±(4.0) | 22.1±(2.9) | 24.0±(3.7) | **21.2±(3.9)**\*\* | **20.4±(3.1)**\* |
| Gisette | 5.5±(0.9) | 5.9±(0.7) | 5.1±(1.3) | 7.1±(1.3) | **4.8±(0.9)**\*\* | **4.2±(0.8)**\* |
| Madelon | 30.8±(3.8) | **15.3±(2.6)**\* | 17.4±(2.6) | **15.9±(2.5)**\*\* | 16.7±(2.7) | 16.6±(2.9) |
| $\#W_1/T_1/L_1$: | 11/4/2 | 10/6/1 | 10/7/0 | 13/2/2 | | |
| $\#W_2/T_2/L_2$: | 9/6/2 | 9/6/2 | 13/3/1 | 12/3/2 | | |

We also plot the average cross-validation error with respect to number of selected features. Fig. 4 shows the two most distinguishable data sets, Semeion and Gisette. We can see that both of our methods, $\mathcal{VMI}_{Naive}$ and $\mathcal{VMI}_{pairwise}$, have lower error rates in these two data sets.

## 5   Related Work

There has been a significant amount of work on information-theoretic feature selection in the past twenty years: [5, 7, 8, 9, 10, 15, 11, 12, 20], to name a few. Most of these methods are based on combinations of so-called *relevant, redundant* and *complimentary* information. Such combinations representing low-order approximations of mutual information are derived from two assumptions, and it has proved unrealistic to expect both assumptions to be true. Inspired by group testing [21], more scalable feature selection methods have been developed, but thos methods also require the calculation of high-dimensional mutual information as a basic scoring function.

Estimating mutual information from data requires a large number of observations—especially when the dimensionality is high. The proposed variational lower bound can be viewed as a way of estimating mutual information between a high-dimensional continuous variable and a discrete variable. Only a few examples exist in literature [22] under this setting. We hope our method will shed light on new ways to estimate mutual information, similar to estimating divergences in [23].

## 6   Conclusion

Feature selection has been a significant endeavor over the past decade. Mutual information gives a general basis for quantifying the informativeness of features. Despite the clarity of mutual information, estimating it can be difficult. While a large number of information-theoretic methods exist, they are rather limited and rely on mutually inconsistent assumptions about underlying data distributions. We introduced a unifying variational mutual information lower bound to address these issues and showed that by auto-regressive decomposition, feature selection can be done in a forward manner by progressively maximizing the lower bound. We also presented two concrete methods using Naive Bayes and pairwise $Q$-distributions, which strongly outperform the existing methods. $\mathcal{VMI}_{naive}$ only assumes a Naive Bayes model, but even this simple model outperforms the existing information-theoretic methods, indicating the effectiveness of our variational information maximization framework. We hope that our framework will inspire new mathematically rigorous algorithms for information-theoretic feature selection, such as optimizing the variational lower bound globally and developing more powerful variational approaches for capturing complex dependencies.

## Footnotes

[1]We omit the results for $MIM$ and $CIFE$ due to space limitations. The complete results are shown in the supplementary Sec. C.

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
