[Supplementary Material]

## Supplementary Material for "Variational Information Maximization for Feature Selection"

## A  Detailed Algorithm for Variational Forward Feature Selection

We describe the detailed algorithm for our approach. We also provide open source code implementing $\mathcal{VMI}_{naive}$ and $\mathcal{VMI}_{pairwise}$ [24].

Concretely, let us suppose class label $\mathbf{y}$ is discrete and has $L$ different values $\{y_1, y_2, ..., y_L\}$; then we define the distribution $q(\mathbf{x}_{S^t}|\mathbf{y})$ vector $Q_t^{(k)}$ of size $L$ for each sample $(\mathbf{x}^{(k)}, \mathbf{y}^{(k)})$ at step $t$:

$$Q_t^{(k)} = \left[ \widehat{q}\left(\mathbf{x}_{S^t}^{(k)}|\mathbf{y} = y_1\right), ..., \widehat{q}\left(\mathbf{x}_{S^t}^{(k)}|\mathbf{y} = y_L\right) \right]^T \tag{20}$$

where $\mathbf{x}_{S^t}^{(k)}$ denotes the sample $\mathbf{x}^{(k)}$ projects onto the $\mathbf{x}_{S^t}$ feature space.

Also, We further denote Y of size $L \times 1$ as the distribution vector of $\mathbf{y}$ as follows:

$$Y = [\widehat{p}(\mathbf{y} = y_1), \widehat{p}(\mathbf{y} = y_2), ..., \widehat{p}(\mathbf{y} = y_L)]^T \tag{21}$$

Then we are able to rewrite $q(\mathbf{x}_{S^{t-1}})$ and $q(\mathbf{x}_{S^{t-1}}|\mathbf{y})$ in terms of $Q_{t-1}^{(k)}, Y$ and substitute them into $\widehat{I}_{LB}(\mathbf{x}_{S^{t-1}} : \mathbf{y})$.

To illustrate, at step $t-1$ we have,

$$\widehat{I}_{LB}\left(\mathbf{x}_{S^{t-1}} : \mathbf{y}\right) = \frac{1}{N} \sum_{\mathbf{x}^{(k)}, \mathbf{y}^{(k)}} \log\left(p\left(\mathbf{x}_{S^{t-1}}^{(k)}|\mathbf{y} = \mathbf{y}^{(k)}\right)\right) - \frac{1}{N} \sum_k \log\left(Y^T Q_{t-1}^{(k)}\right) \tag{22}$$

To select a feature $i$ at step $t$, let us define the conditional distribution vector $C_{i,t-1}^{(k)}$ for each feature $i \notin S^{t-1}$ and each sample $(\mathbf{x}^{(k)}, \mathbf{y}^{(k)})$, i.e.,

$$C_{i,t-1}^{(k)} = \left[ q\left(\mathbf{x}_i^{(k)}|\mathbf{x}_{S^{t-1}}^{(k)}, \mathbf{y} = y_1\right), ..., q\left(\mathbf{x}_i^{(k)}|\mathbf{x}_{S^{t-1}}^{(k)}, \mathbf{y} = y_L\right) \right]^T \tag{23}$$

At step $t$, we use $C_{i,t-1}^{(k)}$ and $Q_{t-1}^{(k)}$ previously stored and get,

$$\begin{aligned}
\widehat{I}_{LB}\left(\mathbf{x}_{S^{t-1} \cup i} : \mathbf{y}\right) &= \frac{1}{N} \sum_{\mathbf{x}^{(k)}, \mathbf{y}^{(k)}} \log\left(p\left(\mathbf{x}_{S^{t-1}}^{(k)}|\mathbf{y} = \mathbf{y}^{(k)}\right) p\left(\mathbf{x}_i^{(k)}|\mathbf{x}_{S^{t-1}}^{(k)}, \mathbf{y} = \mathbf{y}^{(k)}\right)\right) \\
&\quad - \frac{1}{N} \sum_k \log\left(Y^T diag\left(Q_{t-1}^{(k)}\right) C_{i,t-1}^{(k)}\right)
\end{aligned} \tag{24}$$

We summarize our detailed implementation in Algorithm 1.

Updating $Q_t^{(k)}$ and $C_{i,t}^{(k)}$ in Algorithm 1 may vary according to different $Q$-distributions. But we can verify that under Naive Bayes $Q$-distribution or pairwise $Q$-distribution, $Q_t^{(k)}$ and $C_{i,t}^{(k)}$ can be obtained recursively from $Q_{t-1}^{(k)}$ and $C_{i,t-1}^{(k)}$ by noticing that $q(\mathbf{x}_i|\mathbf{x}_{S^t}, \mathbf{y}) = p(\mathbf{x}_i|\mathbf{y})$ for Naive Bayes $Q$-distribution and $q(\mathbf{x}_i|\mathbf{x}_{S^t}, \mathbf{y}) = \left(p(\mathbf{x}_i|\mathbf{x}_{f_t}, y) q(\mathbf{x}_i|\mathbf{x}_{S^{t-1}}, \mathbf{y})^{t-1}\right)^t$ for pairwise $Q$-distribution.

Let us denote $N$ as number of samples, $D$ as total number of features, $T$ as number of selected features and $L$ as number of distinct values in class variable $\mathbf{y}$. The computational complexity of Algorithm 1 involves calculating the lower bound for each feature $i$ at every step which is $O(NDL)$; updating $C_{i,t}^{(k)}$ would cost $O(NDL)$ for pairwise $Q$-distribution and $O(1)$ for Naive Bayes $Q$-distribution; updating $Q_t^{(k)}$ would cost $O(NDL)$. We need to select $T$ features, therefore the time complexity is $O(NDT)$.[2]

**Algorithm 1** Variational Forward Feature Selection (VMI)

---

**Data:** $\left(\mathbf{x}^{(1)}, \mathbf{y}^{(1)}\right), \left(\mathbf{x}^{(2)}, \mathbf{y}^{(2)}\right), ..., \left(\mathbf{x}^{(N)}, \mathbf{y}^{(N)}\right)$
**Input:** $T \leftarrow$ {number of features to select}
**Output:** $F \leftarrow$ {final selected feature set}
$F \leftarrow \{\varnothing\}$; $S^0 \leftarrow \{\varnothing\}$; $t \leftarrow 1$
Initialize $Q_0^{(k)}$ and $C_{i,0}^{(k)}$ for any feature $i$; calculate $Y$
**while** $|F| < T$ **do**
    $\widehat{I}_{LB}\left(\mathbf{x}_{S^{t-1} \cup i} : \mathbf{y}\right) \leftarrow$ {Eq. 24 for each $i$ not in $F$}
    $f_t \leftarrow \underset{i \notin S^{t-1}}{\arg\max}\ \widehat{I}_{LB}\left(\mathbf{x}_{i \cup S^{t-1}} : \mathbf{y}\right)$
    **if** $\widehat{I}_{LB}\left(\mathbf{x}_{S^{t-1} \cup f_t} : \mathbf{y}\right) \leq \widehat{I}_{LB}\left(\mathbf{x}_{S^{t-1}} : \mathbf{y}\right)$ **then**
        Clear $S$; Set $t \leftarrow 1$
    **else**
        $F \leftarrow F \cup f_t$
        $S^t \leftarrow S^{t-1} \cup f_t$
        Update $Q_t^{(k)}$ and $C_{i,t}^{(k)}$
        $t \leftarrow t + 1$
    **end if**
**end while**

---

# B  Optimality Under Tree Graphical Models

**Theorem B.1** (Optimal Feature Selection). *If data is generated according to tree graphical models, where the class label $\mathbf{y}$ is the root node, denote the child nodes set in the first layer as $\mathcal{L}_1 = \{\mathbf{x}_1, \mathbf{x}_2, ..., \mathbf{x}_{L_1}\}$, as shown in Fig. B.1. Then there must exist a step $T > 0$ such that the following three conditions hold by using $\mathcal{VMI}_{naive}$ or $\mathcal{VMI}_{pairwise}$:*

*Condition I: The selected feature set $S^T \subset \mathcal{L}_1$.*

*Condition II: $I_{LB}(\mathbf{x}_{S^t} : \mathbf{y}) = I(\mathbf{x}_{S^t} : \mathbf{y})$ for $1 \leq t \leq T$.*

*Condition III: $I_{LB}(\mathbf{x}_{S^T} : \mathbf{y}) = I(\mathbf{x} : \mathbf{y})$.*

Figure B.1: Demonstration of tree graphical model, label $\mathbf{y}$ is the root node.

*Proof.* We prove this theorem by induction. For tree graphical model when selecting the first layer features, $\mathcal{VMI}_{naive}$ and $\mathcal{VMI}_{pairwise}$ are mathematically equal, therefore we only prove $\mathcal{VMI}_{naive}$ case and $\mathcal{VMI}_{pairwise}$ follows the same proof.

1) At step $t = 1$, for each feature $i$, we have,

$$I_{LB}\left(\mathbf{x}_i : \mathbf{y}\right) = \left\langle \ln\left(\frac{q\left(\mathbf{x}_i|\mathbf{y}\right)}{q\left(\mathbf{x}_i\right)}\right)\right\rangle_{p(\mathbf{x},\mathbf{y})}$$

$$= \left\langle \ln\left(\frac{p\left(\mathbf{x}_i|\mathbf{y}\right)}{\sum\limits_{\mathbf{y}'} p\left(\mathbf{y}'\right) p\left(\mathbf{x}_i|\mathbf{y}'\right)}\right)\right\rangle_{p(\mathbf{x},\mathbf{y})} \tag{25}$$

$$= \left\langle \ln\left(\frac{p\left(\mathbf{x}_i|\mathbf{y}\right)}{p\left(\mathbf{x}_i\right)}\right)\right\rangle_{p(\mathbf{x},\mathbf{y})} = I\left(\mathbf{x}_i : \mathbf{y}\right)$$

Thus, we are choosing a feature that has the maximum mutual information with $\mathbf{y}$ at the very first step. Based on the data processing inequality, we have $I(\mathbf{x}_i : \mathbf{y}) \geq I(desc(\mathbf{x}_i) : \mathbf{y})$ for any $\mathbf{x}_i$ in layer 1 where $desc(\mathbf{x}_i)$ represents any descendant of $\mathbf{x}_i$. Thus, we always select features among the nodes of the first layer at step $t = 1$ without loss of generality. If node $\mathbf{x}_j$ that is not in the first layer is selected at step $t = 1$, denote $ances(\mathbf{x}_j)$ as $\mathbf{x}_j$'s ancestor in layer 1, then $I(\mathbf{x}_j : \mathbf{y}) = I(ances(\mathbf{x}_j) : \mathbf{y})$ which means that the information is not lost from $ances(\mathbf{x}_j) \rightarrow \mathbf{x}_j$. In this case, one can always switch $ances(\mathbf{x}_j)$ with $\mathbf{x}_j$ and let $\mathbf{x}_j$ be in the first layer, which does not conflict with the model assumption.

Therefore, condition I and II are satisfied in step $t = 1$.

2) Assuming condition I and II are satisfied in step $t$, then we have the following argument in step $t + 1$:

We discuss the candidate nodes in three classes, and argue that nodes in ***Remaining-Layer 1 Class*** are always being selected.

***Redundant Class*** For any descendant $desc(S^t)$ of selected feature set $S^t$, we have,

$$I\left(\mathbf{x}_{S^t \cup desc(S^t)} : \mathbf{y}\right) = I\left(\mathbf{x}_{S^t} : \mathbf{y}\right) = I_{LB}\left(\mathbf{x}_{S^t} : \mathbf{y}\right) \tag{26}$$

Eq. 26 comes from the fact that the $desc(S^t)$ carries no additional information about $\mathbf{y}$ other than $S^t$. The second equality is by induction.

Based on Eq. 12 and 26, we have,

$$I_{LB}\left(\mathbf{x}_{S^t \cup desc(S^t)} : \mathbf{y}\right) < I\left(\mathbf{x}_{S^t \cup desc(S^t)} : \mathbf{y}\right)$$
$$= I\left(\mathbf{x}_{S^t} : \mathbf{y}\right) \tag{27}$$

We assume here that the LHS is *strictly* less than RHS in Eq. 27 without loss of generality. This is because if the equality holds, we have $p\left(\mathbf{x}_{S^t}|\mathbf{y}\right) p\left(desc\left(S^t\right)|\mathbf{y}\right) = p\left(\mathbf{x}^t, desc\left(S^t\right)|\mathbf{y}\right)$ due to Theorem 3.1. In this case, we can always rearrange $desc(S^t)$ to the first layer, which does not conflict with the model assumption.

Note that by combining Eqs. 26 and 27, we can also get

$$I_{LB}\left(\mathbf{x}_{S^t \cup desc(S^t)} : \mathbf{y}\right) < I_{LB}\left(\mathbf{x}_{S^t} : \mathbf{y}\right) \tag{28}$$

Eq. 28 means that adding a feature in *Redundant Class* will actually *decrease* the value of lower bound $I_{LB}$.

***Remaining-Layer1 Class*** For any other unselected node $j$ of the first layer, i.e., $j \in \mathcal{L}_1 \backslash S^t$, we have

$$I\left(\mathbf{x}_{S^t} : \mathbf{y}\right) \leq I\left(\mathbf{x}_{S^t \cup j} : \mathbf{y}\right) = I_{LB}\left(\mathbf{x}_{S^t \cup j} : \mathbf{y}\right) \tag{29}$$

The inequality in Eq. 29 is obvious which comes from the data processing inequality [6]. And the equality in Eq. 29 comes directly from Theorem 3.1.

***Descendants-of-Remaining-Layer1 Class*** For any node $desc(j)$ that is the descendant of $j$ where $j \in \mathcal{L}_1 \backslash S^t$, we have,

$$I_{LB}\left(\mathbf{x}_{S^t \cup desc(j)} : \mathbf{y}\right) \leq I\left(\mathbf{x}_{S^t \cup desc(j)} : \mathbf{y}\right)$$
$$I\left(\mathbf{x}_{S^t \cup desc(j)} : \mathbf{y}\right) \leq I\left(\mathbf{x}_{S^t \cup j} : \mathbf{y}\right) \tag{30}$$

The second inequality of Ineq. 30 also comes from data processing inequality.

Combining Eqs. 27 and 29, we get,

$$I_{LB}\left(\mathbf{x}_{S^t \cup desc(S^t)} : \mathbf{y}\right) < I_{LB}\left(\mathbf{x}_{S^t \cup j} : \mathbf{y}\right) \tag{31}$$

Combining Eqs. 29 and 30, we get,

$$I_{LB}\left(\mathbf{x}_{S^t \cup desc(j)} : \mathbf{y}\right) \le I_{LB}\left(\mathbf{x}_{S^t \cup j} : \mathbf{y}\right) \tag{32}$$

Ineq. 31 essentially tells us the forward feature selection will always choose *Remaining-Layer1 Class* other than *Redundant Class*.

Ineq. 32 is saying we are choosing *Remaining-Layer1 Class* other than *Descendants-of-Remaining-Layer1 Class* without loss of generality (for the equality concern, we can have the same argument in step $t = 1$).

Considering Ineqs. 31 and 32, in step $t + 1$, the algorithm chooses node $j$ in *Remaining-Layer1 Class*, i.e., $j \in \mathcal{L}_1 \backslash S^t$.

Therefore, condition I and II hold at step $t + 1$.

At step $t + 1$, if $I_{LB}\left(\mathbf{x}_{S^t \cup j} : \mathbf{y}\right) = I_{LB}\left(\mathbf{x}_{S^t} : \mathbf{y}\right)$ for any $j \in \mathcal{L}_1 \backslash S^t$, that means $I\left(\mathbf{x}_{S^t \cup j} : \mathbf{y}\right) = I\left(\mathbf{x}_{S^t} : \mathbf{y}\right)$. Then we have,

$$I\left(\mathbf{x}_{S^t} : \mathbf{y}\right) = I\left(\mathbf{x}_{\mathcal{L}_1} : \mathbf{y}\right) = I\left(\mathbf{x} : \mathbf{y}\right) \tag{33}$$

The first equality in Eq. 33 holds because adding any $j$ in $\mathcal{L}_1 \backslash S^t$ will not increase the mutual information. The second equality is due to the data processing inequality under tree graphical model assumption.

Therefore, if $I_{LB}\left(\mathbf{x}_{S^t \cup j} : \mathbf{y}\right) = I_{LB}\left(\mathbf{x}_{S^t} : \mathbf{y}\right)$ for any $j \in \mathcal{L}_1 \backslash S^t$, we set $T = t$. Thus by combining condition II and Eq. 33, we have,

$$I_{LB}\left(\mathbf{x}_{S^T} : \mathbf{y}\right) = I\left(\mathbf{x}_{S^T} : \mathbf{y}\right) = I\left(\mathbf{x} : \mathbf{y}\right) \tag{34}$$

Then condition III holds.

$\square$

## C  Datasets and Results

Table 4 summarizes the datasets used in the experiment. Table 5 shows the complete results.

Table 4: **Dataset summary.** $N$**: # samples,** $d$**: # features,** $L$**: # classes.**

| Data | $N$ | $d$ | $L$ | Source |
|---|---|---|---|---|
| Lung | 73 | 325 | 20 | [25] |
| Colon | 62 | 2000 | 2 | [25] |
| Leukemia | 72 | 7070 | 2 | [25] |
| Lymphoma | 96 | 4026 | 9 | [25] |
| Splice | 3175 | 60 | 3 | [26] |
| Landsat | 6435 | 36 | 6 | [26] |
| Waveform | 5000 | 40 | 3 | [26] |
| KrVsKp | 3196 | 36 | 2 | [26] |
| Ionosphere | 351 | 34 | 2 | [26] |
| Semeion | 1593 | 256 | 10 | [26] |
| Multifeat. | 2000 | 649 | 10 | [26] |
| Optdigits | 3823 | 64 | 10 | [26] |
| Musk2 | 6598 | 166 | 2 | [26] |
| Spambase | 4601 | 57 | 2 | [26] |
| Promoter | 106 | 57 | 2 | [26] |
| Gisette | 6000 | 5000 | 2 | [4] |
| Madelon | 2000 | 500 | 2 | [4] |

| Dataset | mRMR | JMI | MIM | CMIM | CIFE | $\mathcal{SPEC}_{\mathcal{CMI}}$ | $\mathcal{VMI}_{naive}$ | $\mathcal{VMI}_{pairwise}$ |
|---|---|---|---|---|---|---|---|---|
| Lung | **10.9±(4.7)**** | 11.6±(4.7) | 18.3±(5.4) | 11.4±(3.0) | 23.3±(5.4) | 11.6±(5.6) | **7.4±(3.6)*** | 14.5±(6.0) |
| Colon | 19.7±(2.6) | 17.3±(3.0) | 22.0±(4.3) | 18.4±(2.6) | 23.5±(4.3) | 16.1±(2.0) | **11.2±(2.7)*** | **11.9±(1.7)**** |
| Leukemia | 0.4±(0.7) | 1.4±(1.2) | 2.5±(1.1) | 1.1±(2.0) | 4.9±(1.9) | 1.8±(1.3) | **0.0±(0.1)*** | **0.2±(0.5)**** |
| Lymphoma | 5.6±(2.8) | 6.6±(2.2) | 13.0±(6.4) | 8.6±(3.3) | 35.6±(4.3) | 12.0±(6.6) | **3.7±(1.9)*** | **5.2±(3.1)**** |
| Splice | **13.6±(0.4)*** | **13.6±(0.5)*** | **13.6±(0.5)**** | 13.7±(0.5) | 14.7±(0.3) | 13.7±(0.5) | 13.7±(0.5) | 13.7±(0.5) |
| Landsat | 19.5±(1.2) | 18.9±(1.0) | 22.0±(3.8) | 19.1±(1.1) | 19.7±(1.7) | 21.0±(3.5) | **18.8±(0.8)*** | **18.8±(1.0)**** |
| Waveform | **15.9±(0.5)*** | **15.9±(0.5)*** | 16.1±(0.8) | 16.0±(0.7) | 22.8±(2.2) | **15.9±(0.6)**** | **15.9±(0.6)**** | **15.9±(0.5)*** |
| KrVsKp | 5.1±(0.7) | 5.2±(0.6) | 5.3±(0.6) | 5.3±(0.5) | **5.0±(0.7)*** | **5.1±(0.6)**** | 5.3±(0.5) | 5.1±(0.7) |
| Ionosphere | 12.8±(0.9) | 16.6±(1.6) | 13.3±(0.9) | 13.1±(0.8) | 16.1±(1.6) | 16.8±(1.6) | **12.7±(1.9)**** | **12.0±(1.0)*** |
| Semeion | 23.4±(6.5) | 24.8±(7.6) | 26.7±(9.7) | 16.3±(4.4) | 28.6±(5.8) | 26.0±(9.3) | **14.0±(4.0)*** | **14.5±(3.9)**** |
| Multifeat. | 4.0±(1.6) | 4.0±(1.6) | 4.9±(2.3) | 3.6±(1.2) | 7.2±(3.0) | 4.8±(3.0) | **3.0±(1.1)*** | **3.5±(1.1)**** |
| Optdigits | 7.6±(3.3) | 7.6±(3.2) | 7.9±(3.9) | **7.5±(3.4)**** | 8.1±(4.2) | 9.2±(6.0) | **7.2±(2.5)*** | 7.6±(3.6) |
| Musk2 | **12.4±(0.7)*** | 12.8±(0.7) | 14.0±(1.2) | 13.0±(1.0) | 13.2±(0.6) | 15.1±(1.8) | 12.8±(0.6) | **12.6±(0.5)**** |
| Spambase | 6.9±(0.7) | 7.0±(0.8) | 7.3±(0.9) | **6.8±(0.7)**** | 10.3±(1.8) | 9.0±(2.3) | **6.6±(0.3)*** | **6.6±(0.3)*** |
| Promoter | 21.5±(2.8) | 22.4±(4.0) | 21.7±(3.1) | 22.1±(2.9) | 27.4±(3.2) | 24.0±(3.7) | **21.2±(3.9)**** | **20.4±(3.1)*** |
| Gisette | 5.5±(0.9) | 5.9±(0.7) | 7.2±(1.2) | 5.1±(1.3) | 6.5±(0.8) | 7.1±(1.3) | **4.8±(0.9)**** | **4.2±(0.8)*** |
| Madelon | 30.8±(3.8) | **15.3±(2.6)**** | 16.8±(2.7) | 17.4±(2.6) | **15.1±(2.7)*** | 15.9±(2.5) | 16.7±(2.7) | 16.6±(2.9) |
| #$W_1/T_1/L_1$: | 11/4/2 | 10/6/1 | 11/6/0 | 10/7/0 | 15/0/2 | 13/2/2 | | |
| #$W_2/T_2/L_2$: | 9/6/2 | 9/6/2 | 15/2/0 | 13/3/1 | 15/1/1 | 12/3/2 | | |

Table 5: **Average cross validation error rate comparison of $\mathcal{VMI}$ against other methods. The last two lines indicate win(W)/tie(T)/loss(L)** for $\mathcal{VMI}_{naive}$ and $\mathcal{VMI}_{pairwise}$ **respectively.**

# D  Generating Synthetic Data

Here is a detailed generating process for synthetic tree graphical model data in the experiment.

Draw $\mathbf{y} \sim Bernoulli(0.5)$

Draw $\mathbf{x}_1 \sim Gaussian(\sigma = 1.0, \mu = \mathbf{y})$

Draw $\mathbf{x}_2 \sim Gaussian(\sigma = 1.0, \mu = \mathbf{y}/1.5)$

Draw $\mathbf{x}_3 \sim Gaussian(\sigma = 1.0, \mu = \mathbf{y}/2.25)$

Draw $\mathbf{x}_4 \sim Gaussian(\sigma = 1.0, \mu = \mathbf{x}_1)$

Draw $\mathbf{x}_5 \sim Gaussian(\sigma = 1.0, \mu = \mathbf{x}_1)$

Draw $\mathbf{x}_6 \sim Gaussian(\sigma = 1.0, \mu = \mathbf{x}_2)$

Draw $\mathbf{x}_7 \sim Gaussian(\sigma = 1.0, \mu = \mathbf{x}_2)$

Draw $\mathbf{x}_8 \sim Gaussian(\sigma = 1.0, \mu = \mathbf{x}_3)$

Draw $\mathbf{x}_9 \sim Gaussian(\sigma = 1.0, \mu = \mathbf{x}_3)$

## Footnotes

[2] We ignore $L$ here because the number of classes is usually much smaller.