[Reviews · NeurIPS 2016]

Reviewer 1

Summary

In feature selection, a natural objective is to find the set of features that maximize the mutual information with the output. This is hard to compute, so you can do a greedy version that successively adds features that add the most information. However, doing so still requires computing the mutual information between one variable and a set of other variables, which takes an exponential amount of data in the number of features chosen. So you need to make some assumptions on the distribution to avoid this issue.

Qualitative Assessment

The argument that existing results require assumptions that are unrealistic, particularly in practice, is pretty convincing. The paper then proposes a different method, which works under less restrictive conditions (naive Bayes) when you know the probability distribution. You need to estimate the probability distribution from the data, and it's not clear how much you lose by doing so. It would be nice to have a measure of how much you lose from q not equalling p. For example, if q and p are close under KL divergence, are you guaranteed to estimate the MI well? The experiments are fairly convincing that the method works well.

Confidence in this Review

2-Confident (read it all; understood it all reasonably well)


Reviewer 2

Summary

This paper proposes a forward sequential approach for feature selection based on approximated mutual information between the outcome and the inputs. The authors start by deriving the criterion to optimize for performing forward feature selection based on mutual information. Then, they review the existing approximations for mutual information, which is required even for medium scale problems. The existing methods basically assume particular factorization of the probability distribution between the features and/or between the features and the outcome in order to reach a cheaper computation of the entropy, and then of the mutual information itself. This paper adopts a different approach, by considering a lower bound for the mutual information based on variational approaches. The authors propose two variational approximations that fit well the forward sequential framework, accompanied with their estimation procedures, thus ending with variational forward feature selection methods. The global complexity of these methods are the same than the ones having stronger assumptions on the distributions of the features. Numerical experiments are made to assess the correctness of the lower bound. Performance comparison are made with existing methods on a series of classical data set.

Qualitative Assessment

pros: the paper is well written and clear. The derivation of the method is clearly stated. There are some theoretical guarantees for the method under certains graphical model on the features. cons: there are relatively few numerical experiments on synthetic data to asses the performances of the method. Considering only forward feature selection (which is not obvious in the title), narrows the potential and applicability of the method. Questions: regarding the cross-validation curves on the real data sets (quite flat), are we in a situation of overfitting or do the curves eventually go up for many selected features? How do the MI based methods compared with non-MI based method on these datasets ?

Confidence in this Review

2-Confident (read it all; understood it all reasonably well)


Reviewer 3

Summary

This paper describes a variational approximation to the mutual information which is used to estimate higher dimensional mutual informations for feature selection. The selection then proceeds by a greedy forward search, with restarts whenever the variational approximation becomes too poor.

Qualitative Assessment

Major comments: There are two aspects to this algorithm which could account for it's improved performance, the variational approximation to the mutual information, or the restarting and clearing of the conditioning set when the approximation becomes poor. This latter aspect isn't discussed much within the paper, but noting the number of times the restart is triggered could help explain the puzzling results (e.g. why the variational approximation with naive bayes outperforms mRMR when it's essentially the same criterion, and why a naive bayes approximation can find useful features in MADELON when the dataset is specifically designed to confound such approximations). A similar search could be designed for mRMR as the selection criterion becomes negative after a certain number of iterations, and at this point the search could be restarted in a similar fashion. I suspect this search has a markedly different behaviour than other searches and so deserves more discussion in the main paper than a single line (line 169). The assumptions from Brown et al 2012 are implicitly invalid as they only hold for a single iteration of a feature selection algorithm, as repeatedly applying Assumption 1 as the feature set grows leads to strange graphical model structures, and the interaction with Assumption 2 makes them invalid. Lines 81-83 touch upon this subject but it could be made more explicit, especially as the variational assumptions remain reasonable through iterations. It would be helpful for the authors to discuss why the naive bayes approximation works so well, and in particular why it seems to outperform the pairwise approximation in many cases. Minor comments: Eq (11) is implied by eq (10), and so is unnecessary. Why not directly calculate the mutual informations in Table 2 rather than estimating them from samples? The paper needs a small amount of grammar cleanup (e.g. line 203 "for the rest datasets" -> "for the rest of the datasets" etc).

Confidence in this Review

3-Expert (read the paper in detail, know the area, quite certain of my opinion)


Reviewer 4

Summary

In this paper, the authors studied the problem of feature selection in machine learning. The authors showed that previous work that selects features by maximizing mutual information approximates the mutual information based on unrealistic assumptions. This paper formulated a more flexible and general class of assumptions based on variational distributions and use them to generate lower bounds for mutual information at a tractable computational cost.

Qualitative Assessment

- In Fig. 3, the gap between the ground truth and the lower bound looks wide from step 3 to step 8. Can the authors explain this? - This paper would benefit from deriving some guarantee on the tightness of the lower bound so as to make this variational method more convincing. - In Section 3.2.1, the authors assumed naive Bayes Q-distribution for their variational method. The authors claimed in the abstract that this assumption was flexible and general. It would be better if the authors could show in the paper that this assumption generalizes a number of existing important models.

Confidence in this Review

2-Confident (read it all; understood it all reasonably well)


Reviewer 5

Summary

The paper proposes an interesting algorithm for feature selection motivated by mutual information maximization. The authors attempt to show problems with current approaches that address the same problem (but I believe that their argument is flawed). The main idea behind their approach is to estimate a lower bound on the mutual information and use this estimate for a greedy feature selection. The paper describes a set of experiments showing clear advantage of the proposed method over other information maximization approaches to feature selection.

Qualitative Assessment

There are several important issues that I believe must be addressed A. The authors make the following argument against current approaches. Current approaches are optimal under a pair of assumptions. These assumptions are hardly ever satisfied simultaneously. Therefore current approaches are flawed. There is a clear flawed logic here. The only thing shown is that current approaches are not optimal. But this is clear even from the greedy perspective. Their algorithms is also clearly non optimal in the general case. I don't think this is such a big deal, except that the authors spend a lot of text with figures, etc., to make this argument. B. I believe the strength of the paper is the new algorithm, and the experimental evaluation which shows its superiority. Still, I would like to point out the following problems with the experiments. B1. They compare the method only to other approaches to information maximization, and not to other feature selection schemes. In particular I was missing comparison to the FS, as described, for example, in the work of Das and Kempe referenced in the paper. B2. The methodology used by the authors (apparently following other work in the field) is to performs the comparison with 3 nearest neighbors and linear SVM. As explained below my concern is mostly with the linear SVM. I was expecting the following two experiments: 1. An estimate of the maximized information showing (hopefully) that the proposed method gives larger values. 2. Nonlinear SVM. Here is why. There is an implied assumption that information maximization is better than correlation based techniques (cf line 24). So One would expect the features selected using information maximization to outperform those selected by, eg, FS. But this would clearly not be the case if the classifier is linear. Example: if both x and x^2 are features, linear classifiers can use both. But information maximization would most likely not select both. This raises the possibility that an algorithm more similar to FS would perform better with linear SVM even though it would produce smaller mutual information. C. I was missing references and discussion of work related to "Markov Blanket". D. Typo in Table 1 caption. What is listed as d should be T. E. The authors state (line 55) that optimal feture selection that maximizes the mutual information is NP-hard. Could you please provide a reference? (I believe the problem is much more difficult.)

Confidence in this Review

2-Confident (read it all; understood it all reasonably well)


Reviewer 6

Summary

In this paper, a new forward feature selection method is proposed by using a lower bound on mutual information. It is proved that under tree graphical models if the data has been generated by Naive Bayes model then the proposed lower bound is exact with uses of Naive Bayes Q-distribution and Pairwise Q-distributions. At last, the new feature selection method is compared with state-of-the-art mutual information based features selection methods on different real-world datasets.

Qualitative Assessment

The novelty and innovation of the paper are low. The assumptions which cause that the lower bound is exact are unrealistic and far from the real-world datasets and there are not any arguments on the gap between the lower bound and exact value of the mutual information. In the other hand, the proposed method does not consider the redundancy of the features in general (some special cases are considered according to the used graphical models). Overall, the approach of the paper which is considering a lower bound to estimate the mutual information is quite interesting but it must theoretically investigate in depth. Despite the theoretical part, extensive experiments have been done which shows that the new method outperforms state-of-the-art mutual information based feature selection methods in practice.

Confidence in this Review

2-Confident (read it all; understood it all reasonably well)